# Targeting of Silver Cations, Silver-Cystine Complexes, Ag Nanoclusters, and Nanoparticles towards SARS-CoV-2 RNA and Recombinant Virion Proteins

**DOI:** 10.3390/v14050902

**Published:** 2022-04-26

**Authors:** Olga V. Morozova, Valentin A. Manuvera, Alexander E. Grishchechkin, Nikolay A. Barinov, Nataliya V. Shevlyagina, Vladimir G. Zhukhovitsky, Vassili N. Lazarev, Dmitry V. Klinov

**Affiliations:** 1Federal Research and Clinical Center of Physical-Chemical Medicine of Federal Medical Biological Agency, 1A Malaya Pirogovskaya St., 119435 Moscow, Russia; vmanuvera@yandex.ru (V.A.M.); nikmipt@gmail.com (N.A.B.); lazar0@mail.ru (V.N.L.); klinov.dmitry@mail.ru (D.V.K.); 2D.I. Ivanovsky Institute of Virology of the National Research Center of Epidemiology and Microbiology of N.F. Gamaleya of the Russian Ministry of Health, 16 Gamaleya St., 123098 Moscow, Russia; grish-sx1@mail.ru (A.E.G.); nvsh@mail.ru (N.V.S.); zhukhovitsky@rambler.ru (V.G.Z.); 3Sirius University of Science and Technology, 1 Olympic Ave., 354349 Sochi, Russia; 4Center for Precision Genome Editing and Genetic Technologies for Biomedicine of Federal Medical Biological Agency, 1A Malaya Pirogovskaya St., 119435 Moscow, Russia; 5Moscow Institute of Physics and Technology, 9 Institutsky Per., 141700 Dolgoprudny, Russia

**Keywords:** nanosilver, beta-coronavirus, RNA-containing bacteriophage MS2, RT^2^-PCR, ELISA, xMAP

## Abstract

**Background**: Nanosilver possesses antiviral, antibacterial, anti-inflammatory, anti-angiogenesis, antiplatelet, and anticancer properties. The development of disinfectants, inactivated vaccines, and combined etiotropic and immunomodulation therapy against respiratory viral infections, including COVID-19, remains urgent. **Aim:** Our goal was to determine the SARS-CoV-2 molecular targets (genomic RNA and the structural virion proteins S and N) for silver-containing nanomaterials. **Methods:** SARS-CoV-2 gene cloning, purification of S2 and N recombinant proteins, viral RNA isolation from patients’ blood samples, reverse transcription with quantitative real-time PCR ((RT)^2^-PCR), ELISA, and multiplex immunofluorescent analysis with magnetic beads (xMAP) for detection of 17 inflammation markers. **Results:** Fluorescent Ag nanoclusters (NCs) less than 2 nm with a few recovered silver atoms, citrate coated Ag nanoparticles (NPs) with diameters of 20–120 nm, and nanoconjugates of 50–150 nm consisting of Ag NPs with different protein envelopes were constructed from AgNO_3_ and analyzed by means of transmission electron microscopy (TEM), atomic force microscopy (AFM), ultraviolet-visible light absorption, and fluorescent spectroscopy. SARS-CoV-2 RNA isolated from COVID-19 patients’ blood samples was completely cleaved with the artificial RNase complex compound Li^+^[Ag^+^_2_Cys_2_^−^(OH^−^)_2_(NH_3_)_2_] (Ag-2S), whereas other Ag-containing materials provided partial RNA degradation only. Treatment of the SARS-CoV-2 S2 and N recombinant antigens with AgNO_3_ and Ag NPs inhibited their binding with specific polyclonal antibodies, as shown by ELISA. Fluorescent Ag NCs with albumin or immunoglobulins, Ag-2S complex, and nanoconjugates of Ag NPs with protein shells had no effect on the interaction between coronavirus recombinant antigens and antibodies. Reduced production of a majority of the 17 inflammation biomarkers after treatment of three human cell lines with nanosilver was demonstrated by xMAP. **Conclusion:** The antiviral properties of the silver nanomaterials against SARS-CoV-2 coronavirus differed. The small-molecular-weight artificial RNase Ag-2S provided exhaustive RNA destruction but could not bind with the SARS-CoV-2 recombinant antigens. On the contrary, Ag^+^ ions and Ag NPs interacted with the SARS-CoV-2 recombinant antigens N and S but were less efficient at performing viral RNA cleavage. One should note that SARS-CoV-2 RNA was more stable than MS2 phage RNA. The isolated RNA of both the MS2 phage and SARS-CoV-2 were more degradable than the MS2 phage and coronavirus particles in patients’ blood, due to the protection with structural proteins. To reduce the risk of the virus resistance, a combined treatment with Ag-2S and Ag NPs could be used. To prevent cytokine storm during the early stages of respiratory infections with RNA-containing viruses, nanoconjugates of Ag NPs with surface proteins could be recommended.

## 1. Introduction

The disinfection of personal protective equipment, production of inactivated vaccines, and treatment of infectious diseases are urgently needed. However, the high mutation rate of RNA-containing viruses, the absence of cellular and viral RNA reparation systems, as well as multiple drug resistance exceed the limited repertoire of currently available antivirals. 

Nanosilver particles with sizes of 1–100 nm in at least one dimension are widely used due to developed construction methods, tunable physicochemical parameters, antiviral, antibacterial, antifungal, anticancer, anti-inflammatory, anti-angiogenesis and antiplatelet properties [1,2,3]. Despite extensive research, the exact mechanisms of nanosilver action remain uncertain. Three models were recently suggested: “Trojan horse”, inductive, and quantum mechanical with possible collaborative effects [3]. According to the most common “Trojan horse” mechanism, Ag nanostructures can serve as an Ag^+^ carriers with permanent release of Ag^+^ ions through oxidative dissolution. Silver NPs exhibit a “core−shell” structure with metallic silver in their central part surrounded by an outer shell of surface oxide or sulfide layers [4]. Replacement of Ag^+^ by electrolyte ions, the potential formation of insoluble AgCl, subsequent catalyzed oxidative corrosion of Ag, and further dissolution of the surface layer of Ag_2_O occur [4,5]. The Ag^+^ cations have an affinity to thiol, amino, phosphate, and carboxyl groups. Ag^+^ containing complexes can interact with DNA [6]. Additionally, the Ag^+^ complexes with anti-inflammatory agents cause DNA fragmentation [3]. Metal ion release, oxidative stress, and non-oxidative mechanisms can occur simultaneously. Silver ions are used as the center of catalytic activity to activate oxygen in air or water, leading to production of reactive oxygen radicals. Ag NPs are known to generate two reactive oxygen species (ROS)-: superoxide (O_2_^∙^^−^) and hydroxyl (OH^∙^) radicals, which attack proteins and depress the activity of enzymes, inhibit the cellular antioxidant defense system, and cause mechanical damages of membranes [1,6]. 

The second mechanism, the inductive mechanism, combines numerous non-oxidative processes such as Ag NPs adhesion to surfaces, electrostatic interaction with membranes, cell destruction, and inappropriate functioning of organelles. Ag^+^ electrostatically interacts with membrane phospholipids and proteins, which can cause depolarization and destabilization of the cellular membrane and the leakage of H^+^. Ag NPs induce disorder at the molecular level of lipid bilayers making them more fluidic and expanded. Ag NPs adhesion can occur through electrostatic attraction or weak interaction forces. Moreover, Ag NPs themselves can disorder the function of different cellular organelles due to adhesion to their surfaces. Furthermore, Ag NPs of ~10 nm can pass through the membrane pores, increasing membrane permeability and inactivating the respiratory chain of electron transfer. In addition, Ag NPs can lead to protein coagulation and change gene expression, in particular for ribosomal proteins and enzymes [3]. Thus, silver nanostructures can induce changes in structures and functions or activate cellular destructive mechanisms; therefore, the suggestion was made to call this mechanism the “inductive” toxicity mechanism [3].

The third mechanism, the quantum mechanical mechanism, includes plasmon modes and quantum states of the metallic NPs: the unique light propagation, the enhanced reactivity of hot electrons, and the relaxation of excited NPs. The hot electrons in noble metal NPs can decay through two different actions, namely electron–electron and electron–phonon interactions. The successful transfer of Ag NPs energy, provided by surface plasmon resonance (SPR) through electron–electron interactions, requires the direct deposition of the substrate molecules on the NPs surface. ROS formation under irradiation was accompanied by a decrease in superoxide dismutase, catalase, and glutathione peroxidase activity. In contrast, electron–phonon interactions lead to thermal diffusion (heat transfer) outside the NPs, which is the basis for photothermal therapy. The NPs are more stable under irradiation and metallic NPs have a significantly higher extinction coefficient due to SPR, which additionally can be tuned by changes in NPs form and size. Ag NPs sizes have a significant impact on their features, which are related to the quantum size effect [3].

Numerous applications of nanosilver recently extended to antivirals [7,8]. Long before pharmaceutical drugs even existed, silver was commonly used to kill more than 650 known pathogens, including the smallpox virus. The adhesion of Ag nanostructures to a cell or virus surface, alterations of membrane properties caused by the formation of free radicals that damage both cellular membranes and viral envelopes, interactions with DNA, and enzyme deterioration play roles in antiviral activity [6]. Additionally, Ag^+^ impasses and destroys both RNA and DNA, thus hindering viral genome replication. Ag^+^ prevents the translation of proteins due to damage to ribosomes. In addition, nanosilver can act as a carrier of other drugs [1]. Thus, disruption of the cellular membranes and viral envelopes, as well as interaction with nucleic acids and proteins [1], are the major known processes of nanosilver-induced disinfecting activity. The multiple mechanisms of nanosilver action against infectious agents require multiple gene mutations for resistance to develop simultaneously; therefore, resistance to Ag-containing materials is hardly possible [1].

Ag NPs exhibit antiviral activity against influenza A virus [8,9,10,11,12], respiratory syncytial virus [13,14], hepatitis B virus [15], monkeypox virus [16], vaccinia virus [17], human parainfluenza virus [18], herpes simplex virus [6,18], and human immunodeficiency virus (HIV) type 1 [19]. Despite extensive research and proven antiviral properties, the implementation of nanosilver in commercial antiviral drugs is limited [2,20,21,22]. Cytotoxicity is a major limitation for their use as antiviral agents in biomedical applications [23]. Commercially available Sovereign Silver and Argentyn 23 manufactured by Natural Immunogenics Corp (NIC) contain a mixture of silver cations and NPs [24]. Currently, colloidal silver remains a certifiable cure without FDA approval (https://www.naturalnews.com/2020-04-01-fda-aggressively-attacks-colloidal-silver-products-coronavirus.html, accessed on 21 April 2022). 

Little is currently known about antiviral properties of silver materials against coronaviruses. The antiviral activity of Ag NCs against a coronavirus porcine epidemic diarrhea virus (PEDV) was assayed [25]. Silver nanoclusters were shown to inhibit viral RNA-dependent RNA replication and PEDV maturation [25]. Ag NPs were studied with similar coronaviruses in vitro and in vivo and the analysis of anti-coronavirus activity, along with toxicological data, suggested that COVID-19 treatment was possible [26]. Ag NPs with diameters of ~10 nm were effective in inhibiting extracellular SARS-CoV-2 at concentrations of 1–10 ppm, while a cytotoxic effect was observed at the concentration of 20 ppm. Ag NPs were shown to inhibit viral entry via disrupting viral integrity [27]. Fluorescent Ag NCs can be also used for coronavirus detection [28]. To our knowledge, neither small-molecular-weight Ag-containing complex compounds nor other artificial RNAses have been analyzed against SARS-CoV-2 isolated RNA and virions. A comparison of interaction of different Ag based materials with SARS-CoV-2 RNA and recombinant proteins has not been described. 

The causative agent of coronavirus disease (COVID-19) is the severe acute respiratory syndrome coronavirus 2 (SARS-CoV-2) [29]. Coronaviruses are spherical enveloped positive single-stranded RNA viruses with a longest single-stranded RNA genome of up to 31 kb in length. The virions consist of structural proteins such as the spike (S), membrane (M), envelope (E), and nucleocapsid (N) proteins. Additionally, there is the hemagglutinin-esterase (HE) protein in some β-coronaviruses [30,31]. The S, M, and E proteins are embedded in the envelope and the N protein interacts with the viral RNA, forming the nucleocapsid [30]. S protein is necessary for β-coronaviruses to attach to the host cell and enter. The host protease furin cleaves the full-length precursor S glycoprotein into two associated polypeptides: S1 and S2 [32]. 

Despite the urgent global need and World Health Organization recommendations, an etiotropic therapy of COVID-19 is currently limited and includes 4 drugs: remdesivir, ritonavir-boosted nirmatrelvir (Paxlovid), sotrovimab, and molnupiravir. None of them contains silver. The main concern is the pheno- and genotypic variability of coronavirus and the consequent resistance to antiviral drugs. Therefore, combined etiotropic therapy is recommended. 

Remdesivir was the first drug, approved by Food and Drug Administration for the treatment of COVID-19 (https://www.covid19treatmentguidelines.nih.gov, accessed on 21 April 2022). Remdesivir is an intravenous adenosine analogue that binds to the viral RNA-dependent RNA polymerase, inhibiting viral RNA replication. 

Three other antiviral agents with emergency use authorization may be used for the treatment of patients with mild to moderate COVID-19 aged ≥ 12 years and weighing ≥ 40 kg who are at high risk of progressing to severe disease. 

Molnupiravir is a synthetic nucleoside derivative of N4-hydroxycytidine that can cause mutations during viral RNA-dependent RNA replication. Molnupiravir’s mutagenic effects could create new variants that evade immunity and prolong the COVID-19 pandemic.

Nirmatrelvir is an orally bioavailable protease inhibitor with antiviral activity against all known human coronaviruses including B.1.1.529 (Omicron). To increase nirmatrelvir concentrations that are effective against SARS-CoV-2, it is packaged with ritonavir (as Paxlovid), a strong cytochrome P450 (CYP) 3A4 inhibitor that has been used to boost HIV protease inhibitors. However, ritonavir may also increase concentrations of certain concomitant medications with possible risk for serious and even life-threatening drug toxicities. 

Immunomodulation of the SARS-CoV-2 induced cytokine storm and inflammation is also necessary. Unspecific immunomodulators approved for the treatment of immune and/or inflammatory syndromes include corticosteroids (e.g., glucocorticoids), interleukins, and kinase inhibitors, as well as interferons (https://www.covid19treatmentguidelines.nih.gov, accessed on 21 April 2022). Convalescent plasma and immunoglobulins may be obtained from patients who have recovered from COVID-19. Additionally, neutralizing monoclonal antibodies directed against SARS-CoV-2 (mainly, the receptor binding domain (RBD) of glycoprotein S1) have been developed. In 2022, FDA issued an emergency use authorization (EUA) for bebtelovimab (recombinant neutralizing human monoclonal antibody that binds to the spike protein of SARS-CoV-2) for the treatment of non-hospitalized patients with mild to moderate COVID-19 who are at high risk of progressing to severe disease. Another monoclonal antibody, Sotrovimab, can be used for treatment of high-risk, non-hospitalized adults with severe combined immunodeficiencies.

The COVID-19 treatment guidelines panel also recommends anticoagulant or antiplatelet therapies, especially for patients with corresponding diagnoses. 

Despite numerous reviews published during the COVID-19 pandemic describing the antiviral potential of silver nanomaterials against SARS-CoV-2, little is currently known about coronavirus molecular targets and especially viral genomic RNA. Virucidal drugs that are capable of destroying viral genome, subgenomic RNA, and surface virion proteins are highly desirable. 

Our research was aimed at SARS-CoV-2 targets for silver containing nanomaterials with immunomodulation properties. 

## 2. Materials and Methods

AgNO_3_ was purchased from Sigma-Aldrich (St. Louis, MO, USA). 

A water-soluble silver (I) complex with cystine Li^+^[Ag^+^_2_Cys_2_^−^(OH^−^)_2_(NH_3_)_2_] (Ag-2S, C_6_H_19_Ag_2_S_2_LiN_4_O_6_, MW = 530.05) was synthesized as previously described [33] and kindly provided by V.N. Silnikov (Institute of Chemical Biology and Fundamental Medicine, Siberian Branch of the Russian Academy of Sciences, Novosibirsk, Russia). In brief, the silver-cystine compound was prepared at room temperature by mixing cystine, lithium hydroxide, silver nitrate, and ammonia in a molar ratio of 1:2:2:8. The reaction mixture was stirred at room temperature for 6 h, and then evaporated at 40–50 °C in vacuum up to 1/4 of the initial volume. After addition of ethanol, the mixture was cooled and incubated at 4–6 °C for 12 h. The resulting water-soluble fine yellow precipitate was filtered, washed with ethanol, and dried. The structure of Ag-2S was confirmed by means of IR, UV, NMR (including ^109^Ag NMR) spectroscopy, and the data of element analysis.

Ag NCs were synthesized using bovine serum albumin (BSA) [34] or immunoglobulins of class G (IgG) with NaBH_4_ recovery. 

Citrate-coated Ag NPs were fabricated by means of Ag^+^ recovery in the presence of sodium citrate and boiling for 1 h [5]. 

Nanoconjugates were fabricated from Ag NPs as described above after 3 washes with deionized water and centrifugation in order to remove citrate anions by means of the nanoprecipitation of proteins from their solutions in fluoroalcohol [5]. 

The structures, sizes, and properties of Ag-2S, Ag-NCs, citrate coated Ag NPs, and nanoconjugates were analyzed by transmission and scanning electron microscopy (TEM and SEM), atomic force microscopy (AFM), dynamic light scattering (DLS), and ultraviolet (UV), visible, and fluorescence spectroscopy. Concentrations of synthesized NPs were measured by means of UV-Vis spectroscopy and atomic absorption spectroscopy (AAS) with corresponding calibration curves. 

RNA-containing phage MS2 was grown as previously described [35] and kindly provided by E.V. Usachev (National Research Center of Epidemiology and Microbiology of Gamaleya, Moscow, Russia). 

### 2.1. Transmission and Scanning Electron Microscopy (TEM and SEM) 

Structures of Ag NCs, NPs, Ag-2S, and nanoconjugates were visualized using transmission electron microscopy (TEM) and atomic force microscopy (AFM) as previously described [5,34]. In brief, Ag NCs with proteins were loaded to “Formvar/Carbon 200 Mesh Copper” copper grids (Ted Pella, Redding, CA 96049-2477, USA) and examined using the TEM system JEM 2100 Plus (JEOL, Akishima-shi, Japan) without contrast. For the Ag NPs conjugated with proteins, additional staining with 1% uranyl acetate was used. 

For elemental analysis, SEM with energy-dispersive X-ray spectroscopy (EDX) was carried out. Silicon wafers were treated in an Electronic Diener plasma cleaner (Plasma Surface Technology, Ebhausen, Germany). Silver nanomaterials were then deposited onto the wafers and characterized using a Zeiss Merlin microscope equipped with GEMINI II Electron Optics (Zeiss, Jena, Germany). The SEM parameters were an accelerating voltage of 1–3 kV and a probe current of 30–80 pA. EDX was performed by SEM via Silicon Drift Detector (SDD) X-MaxN 150 (Oxford Instruments, Abingdon, Oxon, UK) and AZtecEnergy EDX Software (Version 3.0).

### 2.2. Atomic Force Microscopy (AFM) 

Aliquots of 10 µL of Ag-2S or Ag NCs with IgG were placed on the freshly cleaved mica surface for 10 s and then dried with a flow of argon. The samples were analyzed using a Ntegra Prima (NT-MDT, Moscow, Russia) atomic force microscope. All AFM observations were performed using high-resolution silicon cantilevers with resonance frequencies from 190 to 325 kHz with an attraction regime of intermittent contact mode at a scan rate of 1 Hz. Free amplitude of the cantilever in the air was in the 1–10 nm range. For standard image processing and presentation, FemtoScan Online (Advanced technologies center, Russia) was used and for height analysis, SPM Image Magic (http://spm-image-magic.software.informer.com, accessed on 21 April 2022) was used.

### 2.3. Ultraviolet (UV)—Visible (Vis) Light Spectroscopy

UV-Vis absorption spectra of Ag-2S, Ag NCs with proteins, Ag NPs, and nanoconjugates of Ag NP with immunoglobulins, albumins, and fibrinogens were obtained using a NanoDrop 2000c UV-Vis spectrophotometer (Thermo Scientific, Waltham, MA, USA).

### 2.4. Fluorescence Spectroscopy

Fluorescence excitation/emission spectra were measured in a quartz cuvette using the FluoroMax+ spectrofluorometer (Horiba Scientific, Kisshoin Minami-Ku Kyoto, Japan).

### 2.5. RNA Isolation

The study was conducted in accordance with the Declaration of Helsinki. Written informed consent was obtained from the included patients regarding the approval to perform the required investigations. Ethical approval from the Ethics Committee of the National Research Center of Epidemiology and Microbiology of N.F. Gamaleya of the Russian Ministry of Health, Moscow, Russia was obtained before starting the research. 

Human blood mononuclear cells (BMC) or sera of patients (42 males and 53 females, mean age 34.4 ± 3.7) with COVID-19 confirmed via PCR of nasopharyngeal swabs were collected in the spring–summer season of 2020 in Moscow, Russia. After addition of lysis solution containing 5.5 M guanidinium isothiocyanate (GITC), the blood samples were used for total RNA isolation or stored at −80 °C for a few days. 

Nucleic acids were isolated from the BMC or sera of patients diagnosed with COVID-19 in Moscow, Russia and from MS2 phage using GITC-mediated lysis with subsequent alcohol precipitation according to instructions of the Proba-NK kit (DNA-technology, Moscow, Russia). 

### 2.6. RNA Cleavage 

The isolated RNA was dissolved in sterile RNase-free water and incubated with AgNO_3_, Ag-2S, Ag NCs, Ag NPs, and nanoconjugates at 37 °C for 2 h. Then samples were centrifuged at maximal speed 14,000 rpm for 10 min to remove Ag NPs and nanoconjugates. The RNAs were again precipitated from supernatants with isopropanol in order to separate small molecular weight substances such as AgNO_3_, Ag-2S, and, partly, Ag NCs. Then, they were washed twice with 70% ethanol, washed once with acetone and dried. 

### 2.7. Reverse Transcription and Real Time PCR

Reverse transcription (RT) was performed with random N6 primer using the Reverta-L kit (AmpliSens, Moscow, Russia). PCR with specific primers SARS-CoV-2-ORF1ab-F: 5′-GGATCAAGAATCCTTTGGTGG-3′ and SARS-CoV-2-ORF1ab-R: 5′-GTCACAAAATCCTTTAGGATTTGGA-3′ as well as fluorescent hydrolysis probe SARS-CoV-2-ORF1ab-Probe: ROX-CATCGTGTTGTCTGTACTGCCGTTGCC-BHQ2 was carried out in the following regime: 95 °C 5 min, 95 °C 10 s, and 60 °C 1 min (45 cycles) [36]. To detect MS2 phage RNA primers, MS2-TM3-F sense 5′-GGCTGCTCGCGGATACCC-3′ (3166–3183) and MS2-TM3-R anti-sense TM3 5′-TGAGGGAATGTGGGAACCG-3′ (3367–3349) [35,37] as well as MS2-TM3-P TaqMan probe ROX-5′-TCACCGACAGCATGAAGTCCGCCGGT-3′-BHQ2 (3289–3313), kindly provided by N.A Kuznetsova (National Research Center of Epidemiology and Microbiology of Gamaleya, Moscow, Russia), were used in PCR (95 °C 5 min and 45 following cycles 95 °C 10 s, 60 °C 25 s, 72 °C 15 s). The quantitation of genome equivalents was based on Lukyanov–Matz’s equation using a calibration curve of standards.

### 2.8. ELISA

SARS-CoV-2 RNA was isolated from the nasopharyngeal swabs of patients with a confirmed diagnosis of COVID-19 in Moscow in Spring 2020 using the TRIzol LS Reagent kit (Thermo Fisher Scientific, Waltham, MA, USA). Then, 10 μL of total RNA were used for RT with RevertAid RT Reverse Transcription kit (Thermo Fisher Scientific, Waltham, MA, USA) and a random N6 primer. DNA fragments encoding the viral antigens were obtained by PCR with the following primer pairs:(1)SARS-CoV-2 N gene full-length coding region:CoVgN-N 5′-GGGATCCTCTGATAATGGACCCCAAAATCA-3′CoVgN-C 5′-ATAGAATTCTTAGTCGACGGCCTGAGTTGAGTCAGCAC-3′;(2)5′-terminal fragment of the SARS-CoV-2 gene S coding N-terminal part of Spike protein (S) without signal peptide (16–685 aa):CoVgS1-N 5′-GGGATCCGTTAATCTTACAACCAGAACTC-3′CoVgS1-C 5′-TCAGGTACCGTCGACACGTGCCCGCCGAGGAGA-3′;(3)3′-terminal fragment of the SARS-CoV-2 gene S encoding C-terminal part of S protein without transmembrane peptide (686–1213 aa):CoVgS2-N 5′-GGGATCCAGTGTAGCTAGTCAATCCATC-3′CoVgS2-C 5′-TCAGGTACCGTCGACTGGCCATTTTATATACTGCTCA-3′.

The purified PCR products were hydrolyzed with the restriction endonucleases BamHI and SalI, and ligated, thus linearizing the pET-22(b) (Novagen, Moscow, Russia) derivative, the expression plasmid pET-min [38]. Cells from the competent *E. coli* strain Top10 were transformed and the recombinant bacterial clones were selected using PCR with the universal primers T7 (5′-TAATACGACTCACTATAGGG-3′) and T7t (5′-GCTAGTTATTGCTCAGCGG-3′). The structures of the recombinant plasmids were confirmed by Sanger sequencing.

The resulting recombinant plasmids pET-CoV2-gN(H), pET-CoVgS16-685(H), and pET-CoVgS686-1213(H) contained the SARS-CoV-2 full-length N gene, S1 gene fragment and S2 gene fragment, respectively, under the T7 promoter control. The corresponding recombinant proteins N, S1, and S2 possessed a C-terminal His6 tag and were purified by affinity chromatography with a Ni Sepharose High Performance column (GE Healthcare, CIIIA, Chicago, IL, USA) from *E. coli* (strain BL21-gold(DE3)) transformed with the recombinant plasmids pET-CoV2-gN(H), pET-CoVgS16-685(H), and pET-CoVgS686-1213(H).

The purified recombinant proteins S1, S2, and N of SARS-CoV-2 were immobilized on highly activated polystyrol plates for immunological assays and blocked with 1% BSA in phosphate buffer solution (PBS) for 2 h at 37 °C or at 4 °C overnight. Then, immune complexes were revealed using secondary antibodies against human IgG conjugated with horseradish peroxidase and subsequent staining with 3,3′,5,5′-tetramethylbenzidine (TMB) with hydrogen peroxide. Recombinant antigens without treatment with the nanosilver were used as positive controls and PBS as a negative control. All experiments were performed in quadruplicate with subsequent averaging of values. 

### 2.9. Cell Cultures 

Human larynx carcinoma HEp-2, oral epithelial carcinoma L41, and colorectal adenocarcinoma HT-29 cells were obtained from the Russian State Tissue Culture Collection (National Research Center of Epidemiology and Microbiology of N.F. Gamaleya, Moscow, Russia) and grown in culture medium 199 (https://paneco-ltd.ru/catalog/pitatelnaya-sreda-199, accessed on 21 April 2022) supplemented with 8% fetal calf serum (FCS) (HyClone, Thermo Scientific, USA) in the presence of 100 U/mL penicillin and 100 U/mL streptomycin at 37 °C and 5% CO_2_ until ~80% confluent monolayers for 36–48 h. 

### 2.10. Cytotoxicity Analysis

Freshly prepared solutions with concentrations of Ag-2S, Ag NCs, Ag NPs, and nanoconjugates in a 1–1000 μg/mL range were added to wells of 48-well polystyrol plates and their toxicity was determined in quadruplicate by means of MTT test with subsequent averaging of values.

MTT test was performed for 3 cell types to evaluate the in vitro cytotoxicity of AgNO_3_, Ag-2S, Ag NCs, Ag NPs, and nanoconjugates. Sterile MTT solution (5 mg/mL) in PBS was added into each well and incubated during optimal time for optical density measurements in a linear range for 4 h. Then, 100 μL of DMSO was pipetted into each well and carefully mixed to solubilize the crystal formazan. The absorbance was measured at 570 nm. Culture media with 8% FCS in the presence of corresponding Ag-containing nanomaterials and MTT without control intact cells were used as alternative blanks.

### 2.11. Multiplex Immunofluorescent Analysis with Magnetic Microspheres

AgNO_3_, Ag-2S, Ag NCs, and the nanoconjugates Ag NP with BSA, IgG, and fibrinogen at different concentrations below their corresponding CC_50_ were added in culture media in wells of polystyrol plates with ~80% confluent monolayers of 3 different human cell cultures HEp-2, L41, and HT-29. After incubation at 37 °C and 5% CO_2_ for 1, 2, 3, or 4 days the inflammation markers were detected in culture media of the human cells by Unknown (X) Multi Analyte Profiling (xMAP) using the 17 plex kit (BioRad, Hercules, CA, USA) according to manufacturer’s instructions.

### 2.12. Statistical Analysis 

Statistical calculations were performed using Microsoft Excel software. A value of *p* < 0.05 was considered statistically significant. 

## 3. Results and Discussion 

Nanosilver may be applied for disinfection, inactivated vaccine production, and combined therapy without a risk of pathogen resistance due to the multiple mechanisms of action [1,3] and allergic complications for hosts because of nanosilver-induced immunosupression [20,21]. Cellular uptake of nanosilver may mainly induce T-helper 1 cellular immune response necessary for elimination of intracellular pathogens. 

### 3.1. Structure and Physico-Chemical Properties of Silver Nanomaterials

There are numerous tools that can be used for characterizing nanoparticles. While it is best to have multiple orthogonal characterization techniques, there is an international consensus for transmission electron microscopy (TEM) being the “gold standard” technique for nanomaterial characterization. The European Food Safety Authority requires two methods to characterize size distribution of nanomaterials in foods, one of which must be TEM and other may be chosen [24]. To corroborate the TEM findings, atomic force microscopy (AFM) showing height images and size distribution histogram has been often employed [24]. 

For visualization of single Ag nanoparticle and characterization of their dimensions, TEM and AFM were used. The obtained TEM (Figure 1) and AFM (Figure 2) images demonstrated significant variations in the morphology and size of Ag-containing nanomaterials. The size range of blue fluorescent Ag NCs with IgG according to TEM images (Figure 1) was 0.6–2 nm and can be distinguished from the surrounding less dense protein stabilizers and numerous aggregates and agglomerates (Figure 1). On the basis of the AFM images, height histograms of Ag-2S and Ag NCs-IgG demonstrated relatively narrow distributions (Figure 2) with the most probable values being ~1 and ~4 nm for Ag-2S and Ag NCs-IgG, respectively. The bigger apparent dimensions of Ag NCs-IgG in AFM images than in TEM images may be rationalized by the fact that the dimensions in AFM images included the protein shells around them.

Surface topography using AFM resulted in overestimation due to protein stabilizers (IgG) with known heights of ~5 nm [39] (Figure 2) and did not permit to determine the real size ranges of the Ag NCs (Figure 1). On the contrary, AFM of the complex compound Ag-2S allowed us to compare the diameters and heights of nanostructures and to reveal spherical molecules of 1–2 nm (Figure 2). Ag NPs and their nanoconjugates with proteins also significantly varied in diameters from 20 to 150 nm (Figure 1) with polydispersity indexes of 0.1–0.2. 

Properties of Ag-2S were confirmed by UV spectroscopy, IR analysis (Appendix A), NMR (including ^109^Ag NMR) spectroscopy, and energy-dispersive X-ray spectroscopy (EDX) element analysis and completely coincided with previously available data [33]. Structure and physico-chemical properties of Ag-2S remained stable during storage in water solution up to 10 years. 

Broad excitation spectra of fluorescent Ag NCs were in the range of 340–540 nm (Appendix A). Different emission spectra correlated with the original AgNO_3_ concentrations used for Ag NCs fabrication, whereas fluorescence emission intensity depended on protein concentrations from 1 to 25 mg/mL (Appendix A). Ag NCs stabilized with IgG (Figure 1 and Figure 2, Appendix A) or albumin (Appendix A) with blue fluorescence and emission maximum at 420 nm contained heterogeneous metal NCs from 0.6 nm. Green Ag NCs with proteins had a bright fluorescence at 430–470 nm and the red NCs showed an emission maximum at 650 nm (Appendix A). TEM revealed discrete Ag NCs within a broad size range (Figure 1) and their numerous aggregates and protein agglomerates in each sample of fluorescent NC (Figure 1), in spite of different fluorescent emission spectra (Appendix A).

Citrate-coated Ag NPs with the Ag_2_O surface layer are not stable in the presence of phosphate buffer solution (PBS) or culture media with electrolyte solutions during 1 h at room temperature due to replacement of Ag^+^ by electrolyte ions, potential formation of insoluble AgCl, subsequent catalyzed oxidative corrosion of Ag, and further dissolution of the surface layer of Ag_2_O as well as NP aggregation [4]. Protein surface shells protected Ag NPs from dissolution and aggregation. Therefore, the nanoconjugates of the noble metal NPs with proteins remained stable for several months at 4 °C [5,20,21]. 

Multiple known mechanisms of action of nanosilver determine its toxicity in vitro and in vivo; therefore, it remains a major concern for biomedical implementations. Cytotoxic concentrations for 50% cells (CC_50_) on the basis of the MTT test slightly differed for three different human tissue cultures (Table 1). 

Protein envelopes of Ag NCs and nanoconjugates did not significantly reduce their toxicity (Table 1), probably due to proteolysis in lysosomes after cellular endocytosis. However, comparison of fluorescent noble metal NC with BSA showed that Ag NCs with CC_50_ ~10 μg/mL (Table 1) were less toxic than Cd and Au NCs [34]. The artificial RNase Ag-2S appeared to be highly toxic for human cells with a CC_50_ varying in a range of 0.01–0.04 mM corresponding to 5.3–21.2 μg/mL (Table 1). Despite the long-term and high stability of Ag-2S for many years, its cytotoxicity may be caused by the unspecific cleavage of cellular RNA [40]. The cytotoxicity of Ag nanoconjugates with IgG essentially exceeded that of AgNO_3_ (Table 1), perhaps due to the cellular uptake of silver nanostructures covered with IgG but not Ag^+^ ions. The subsequent “Trojan horse” results in high Ag^+^ intracellular concentration and interaction with proteins, nucleic acids, and membranes [1,3,20,21]. 

RNase and antigen binding properties of AgNO_3_, Ag-2S, fluorescent Ag NCs with BSA and immunoglobulins, and citrate coated Ag NPs and their nanoconjugates with protein shells were analyzed by means of RT^2^-PCR and ELISA. 

### 3.2. RNA Cleavage

Screening of RNase activity of the nanosilver was performed with two viral RNA isolated from bacteriophage MS2 and COVID-19 patients’ blood samples as well as in whole MS2 phages and SARS-CoV-2 virus particles (Figure 3). 

#### 3.2.1. RNase Properties of Ag-2S Complex

Complete cleavage of total RNA after incubation with 2 mM Ag-2S for 1 h at 37 °C was earlier shown for influenza A viral and cellular RNA [40]. Both MS2 and SARS-CoV-2 RNA appeared to be more stable in virions and human blood, respectively, than isolated pure RNA (Table 2) due to the poor penetration of Ag-2S into virions and cells [40].

Artificial RNase Ag-2S destroyed viral (Table 2) and cellular RNA [40]. However, neither cellular DNA nor viral antigens were damaged after incubation with Ag-2S [40].

#### 3.2.2. Partial RNA Cleavage with AgNO_3_ and Citrate Coated Ag NPs

Dynamics of MS2 phage and coronaviral RNA degradation in the presence of 0.1 mg/mL Ag-containing nanomaterials exceeded the CC_50_ of the majority of the studied compounds including the Ag-2S complex, Ag NCs, Ag NPs, and nanoconjugates with the only exception AgNO_3_ (CC_50_ 100 μg/mL = 0.1 mg/mL for L41 and HT-29 cells) (Table 1) was assayed by RT^2^-PCR after 15, 30, 60, 90, and 120 min incubation at 37 °C. Thus, isolated MS2 RNA was cleaved in 15 min of treatment with 0.1 mg/mL AgNO_3_ (Table 2), treatment of MS2 phage under similar conditions resulted in a four times lower amount of phage RNA (ΔCt = 2.0) (Table 2) whereas incubation for at least 1 h at 37 °C was required for complete damage of RNA inside the MS2 phage (data not shown). Complete cleavage of SARS-CoV-2 RNA from the COVID-19 patient blood samples was not found (Table 2) perhaps because of the protection of nucleocapsid (N), membrane (M) and spike (S) virion structural proteins, binding with blood proteins, membrane vesicles or cellular debris.

One should note that isolated MS2 RNA complete cleavage took place in 15 min at 37 °C in the presence of 0.1 mg/mL AgNO_3_ and Ag-2S, whereas incubation with the same Ag-containing materials for 1 h at 37 °C was necessary for exhaustive hydrolysis of SARS-CoV-2 RNA (Table 2) as well as for influenza A viral and cellular RNA [40]. 

Partial RNA cleavage in the presence of Ag NPs (Table 2) can be caused by the NPs’ dissolution in the presence of polyanion RNA with Ag^+^ release. Silver ions are known to destroy nucleic acids ([20,21] and references therein). 

#### 3.2.3. Fluorescent Ag NCs and Ag NPs Covered with BSA and IgG without RNase Activity

Incubation of SARS-CoV-2 RNA with Ag NCs (Figure 1 and Figure 2, Appendix A) and nanoconjugates of Ag NPs with immunoglobulins (Figure 1 and Figure 3) or fibrinogens did not cause any significant shifts (ΔCt = Ct (sample) − Ct (control with water)) in the RT^2^-PCR data (Table 2). Consequently, the protein shells of Ag NCs and nanoconjugates of Ag NPs with the major blood proteins (albumin, immunoglobulins, and fibrinogen) prevented RNA cleavage by the nanosilver. However, nanoconjugates with BSA could slightly hydrolyze isolated viral RNA (ΔCt = 1.3–3.7), probably due to the RNase activity of albumin itself [41]. 

### 3.3. Impairments of SARS-CoV-2 S2 and N Antigenic Structures

SARS-CoV-2 virions consist of surface spike S protein embedded in a membrane lipid bilayer and a nucleocapsid containing N protein and genomic RNA [30,31,32]. Both the membrane envelope and structural proteins protect genomic RNA and serve as additional molecular targets for nanosilver. 

Antigenic structures of SARS-CoV-2 recombinant antigens were assayed by ELISA with the virus-specific polyclonal antibodies from COVID-19 patients’ blood sera. Both AgNO_3_ and Ag NPs could inhibit binding of recombinant proteins S2 and N with human polyclonal antibodies (Figure 4) in a clear dose-dependent manner (Figure 5). Notably, the lowest concentration of AgNO_3_ for significant reproducible binding suppression of both coronavirus proteins was 100 μg/mL (0.01%) and ~5 μg/mL (corresponding to 0.1 o.u./mL) for Ag NPs (Figure 5). These ELISA inhibition concentrations were higher cytotoxic concentrations CC_50_ (Table 1) and can be used for disinfectants or preparation of inactivated vaccines but not for therapy. 

The SARS-CoV-2 N and S2 proteins might recover Ag^+^ cations as other proteins [42,43] with possible formation of Ag NCs and inevitable conformational changes of original proteins [34] with antigenic structure deformations. On the other hand, citrate coated Ag NPs are capable to bind with a majority of proteins resulting in nanoconjugates with silver cores and protein shells [5]. Protein localization in close proximity to the Ag NPs surface might change antigenic determinants and hamper their binding with specific antibodies due to steric hindrances.

However, the ELISA data in the presence of Ag-2S, Ag NCs and nanoconjugates with BSA, fibrinogen and hIgG did not significantly differ from the positive controls in the presence of water (Figure 4), probably because of the protein shelters of the silver nanostructures. Therefore, the correlation with concentrations of these Ag-containing nanomaterials was not analyzed. 

### 3.4. Anti-Inflammation Properties of the Nanosilver

Besides cytotoxicity, interaction of cells with the nanosilver can induce innate immunity. Cellular uptake, subsequent stages of endocytosis, proteolytic hydrolysis of protein shells in lysosomes, and antigen presentation with major histocompatibility complex (MHC) type I or II may result in unspecific resistance, cytokine storm, inflammation, adaptive immunity, and allergic complications dependent on cytokine production immediately after interaction with cells. To determine their inherent immunogenicity, Ag-containing nanomaterials, lacking any known immunostimulatory component, were added to three human tissue cultures: HEp-2, L41, and HT-29. Cytokine gene expression was measured by RT^2^-PCR, ELISA (data not shown), and xMAP in dynamics each day during a week. The maximal production of all inflammation biomarkers in HEp-2, L41, and HT29 human cells was observed during the first two days post treatment. Multiple analyte profiling (xMAP) with fluorescent magnetic beads of 17 key inflammation biomarkers including T-helper (Th)1 cytokines: interferon (IFN) γ, tumour necrosis factor (TNF) α, interleukin (IL) 1β, IL12 (p70); Th2 cytokines: IL2, 4, 5, 6, 7, 8, 10, 13; Th17–IL17A; and other inflammation biomarkers: granulocyte colony-stimulating factor (G-CSF), granulocyte macrophage colony stimulating factor (GM-CSF), monocyte chemotactic protein 1 (monocyte chemotactic and activating factor) (MCP-1 (MCAF)), macrophage inflammatory protein 1β (MIP-1β) was the most informative approach, and was necessary to determine the polarization indexes of immune response, pro-inflammation, and potential allergic properties of the nanosilver. 

Two days after the addition of Ag-containing materials to HEp-2 human cells, a significant enhancement in IL8 production (up to 0.94 pg/mL, or 3.7 times) was found. Smaller increases, 2.11 and 1.38 times (up to 0.08 pg/mL) for IL1β in the presence of Ag NPs-IgG and AgNO_3_, respectively, were revealed. Downregulation of IL10 also occurred (below the IL-10 production in control cells, with a concentration of less than 0.05 pg/mL), whereas production of other 13 biomarkers did not essentially change (Table 3). 

IL production is known to depend on the origin of human cells. Thus, IL1β is mainly produced by macrophages and monocytes and is responsible for the regulation of inflammation and the stimulation of acute phase IL2, IL3, IL6, and TNFα as well as temperature growth and fever. Therefore, inflammation is hardly possible with a low level of IL1β and stimulated cytokines. The only exception was IL1β-induced upregulation of IL8 (Table 3), which is associated with acute and chronic inflammation. Silver ions at a concentration 10 μg/mL lower than the CC_50_(Ag^+^) = 31.75 μg/mL (corresponding CC_50_(AgNO_3_) = 50 μg/mL) for the most sensitive cells HEp-2 (Table 1) were not toxic for three human cell lines and could not penetrate through membranes in the cells. Therefore, the observed immunomodulation with AgNO_3_ was modest, if at all. 

Immunomodulation with nanoconjugates of Ag NPs covered with albumins, fibrinogens, or immunoglobulins differed (Table 3). A significant increase in IL1β production (2.11 times higher—up to 0.12 pg/mL) was detected in two days after the treatment of HEp-2 cells with Ag NPs-IgG simultaneously and in a similar manner to its increase after AgNO_3_ addition. One of possible reasons for this is the occurrence of proteolytic hydrolysis of the surface proteins in lysosomes with subsequent MHC I antigen presentation and T-helper I (Th1) cytokine gene expression but not the influence of the silver core. However, the IL1β enhancement did not stimulate the production of other cytokines (Table 3). The growth of MCP-1 (MCAP) secretion (up to 0.37 pg/mL) was caused by Ag NPs-fibrinogen. The inhibition of IL8 and regulatory IL10 production in the presence of all nanoconjugates with Ag core and protein shells resulted in slight changes (if any) in the levels of 17 biomarkers. 

Immunosuppression patterns were similar for all three human immortal cell lines but further research with primary immune cells including macrophages and neutrophils is required. 

Taken together, our data on the antiviral and immunomodulatory properties of nanosilver materials did not confirm the previous reports for other coronavirus species, such as porcine epidemic diarrhea virus (PEDV) [25]. Ag NCs with various proteins and different fluorescence emission spectra (Appendix A) remained inefficient in both SARS-CoV-2 RNA cleavage and virion protein epitope damage (Figure 3 and Figure 4, Table 2), perhaps due to protein shielding. Moreover, positive upregulation of cytokine gene expression and secretion was not found with a few exceptions for IL1β and IL8 production (Table 3). All silver nanomaterials revealed anti-inflammation features that can be implemented against cytokine storm. Evidently, none of the above Ag-based materials may be used as adjuvants to increase immune response. 

## 4. Conclusions

Nanomaterials differ from conventional small-molecule drugs used for the treatment of infectious disease. The sizes, absorption and fluorescence spectra, antiviral and immunomodulation properties of the silver nanostructures also varied markedly from each other. The only low-molecular-weight artificial RNase—the Ag-2S complex—provided complete cleavage of viral and cellular RNA but did not interact with coronavirus recombinant proteins. On the contrary, AgNO_3_ and citrate-coated Ag NPs impaired the binding of SARS-CoV-2 recombinant antigens S2 and N with virus-specific polyclonal antibodies in dose-dependent manner. It is noteworthy that fluorescent Ag NCs and nanoconjugates of Ag NPs with albumin, fibrinogen and immunoglobulins did not deteriorate either viral genomic RNA or antigenic structures, probably due to surface protein shells. The antiviral activity of the studied Ag-containing materials. including both viral RNA cleavage and antigen-antibody binding. was found at concentrations exceeding cytotoxic concentrations. Moreover, the nanosilver suppressed the production of a majority of inflammation biomarkers. Taken together, the silver-containing materials could be used for disinfection, inactivated vaccine production, as well as for combined etiotropic and immunomodulation therapy after local but not systemic administration.

## Figures and Tables

**Figure 1 viruses-14-00902-f001:**
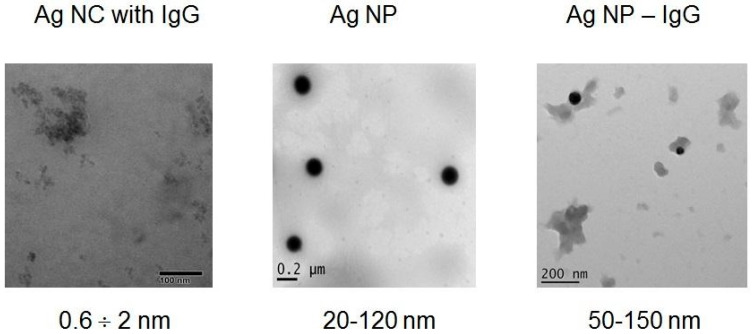
TEM images with size ranges for Ag NCs with IgG, citrate coated Ag NPs and nanoconjugates of Ag NPs with IgG.

**Figure 2 viruses-14-00902-f002:**
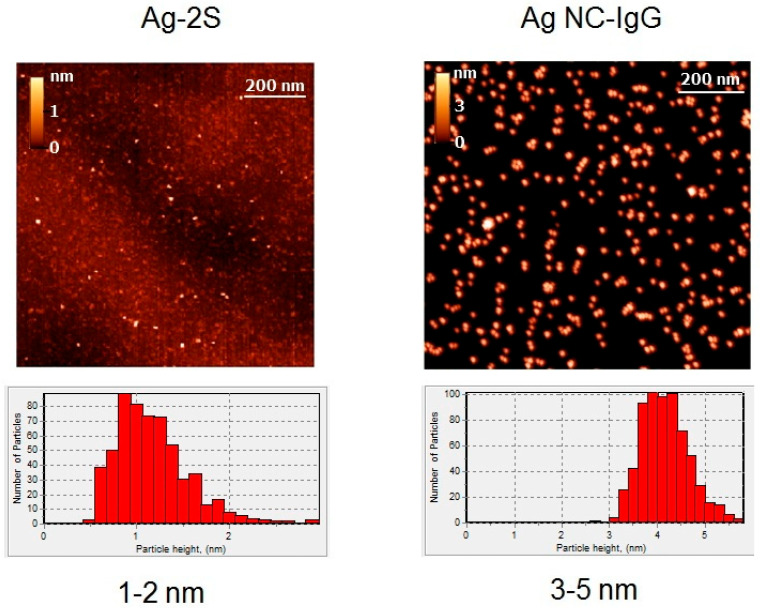
AFM images and height distribution histograms of Ag-2S and Ag NCs with IgG.

**Figure 3 viruses-14-00902-f003:**
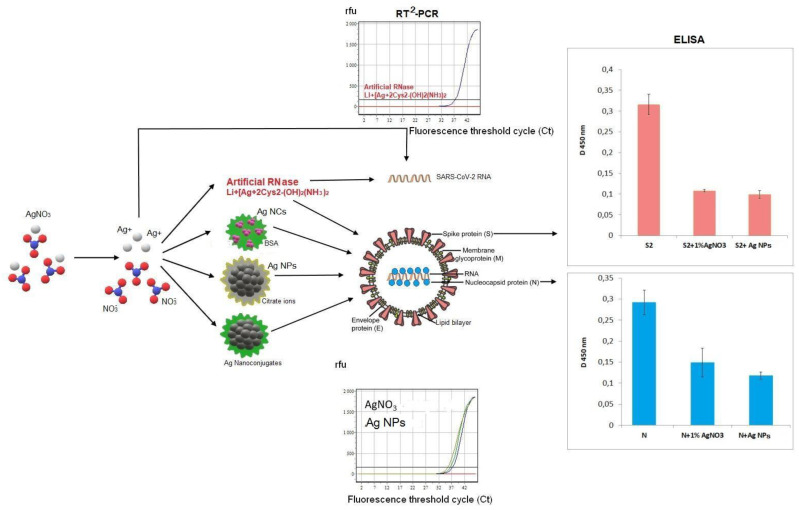
Scheme of SARS-CoV-2 molecular targets and part of available data of RT^2^-PCR and ELISA. SARS-CoV-2 spike surface protein S is shown in pink in both central virion scheme and in the corresponding ELISA data (the histogram in the upper right corner); the coronavirus nucleocapside protein N is shown in blue colour in the central scheme and in the corresponding ELISA data (the histogram in the lower right corner). RT^2^-PCR data demonstrate the fluorescent curves of dependence between PCR cycles (axis X with threshold cycle (Ct)) and the fluorescence emission in relative fluorescence units (r.f.u.).

**Figure 4 viruses-14-00902-f004:**
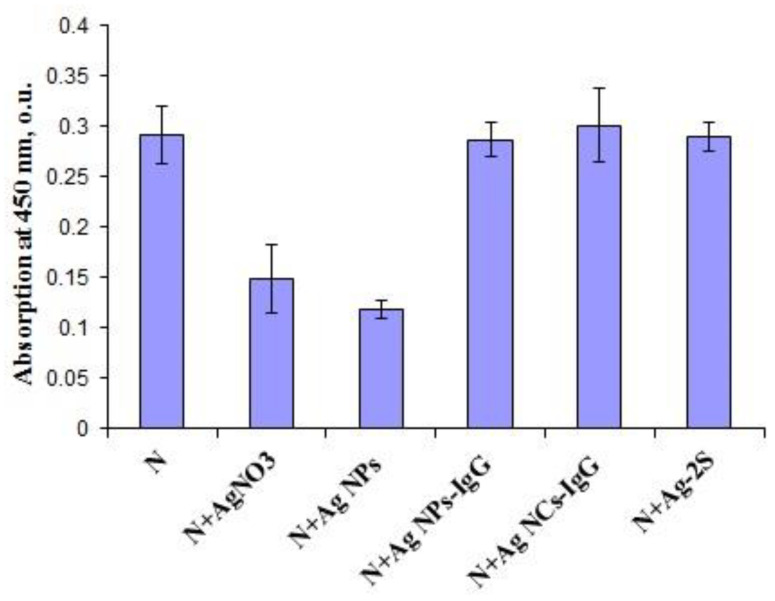
ELISA results for SARS-CoV-2 recombinant antigen N after incubation with AgNO_3_, Ag NPs, Ag NPs-IgG, Ag NCs-IgG and Ag-2S.

**Figure 5 viruses-14-00902-f005:**
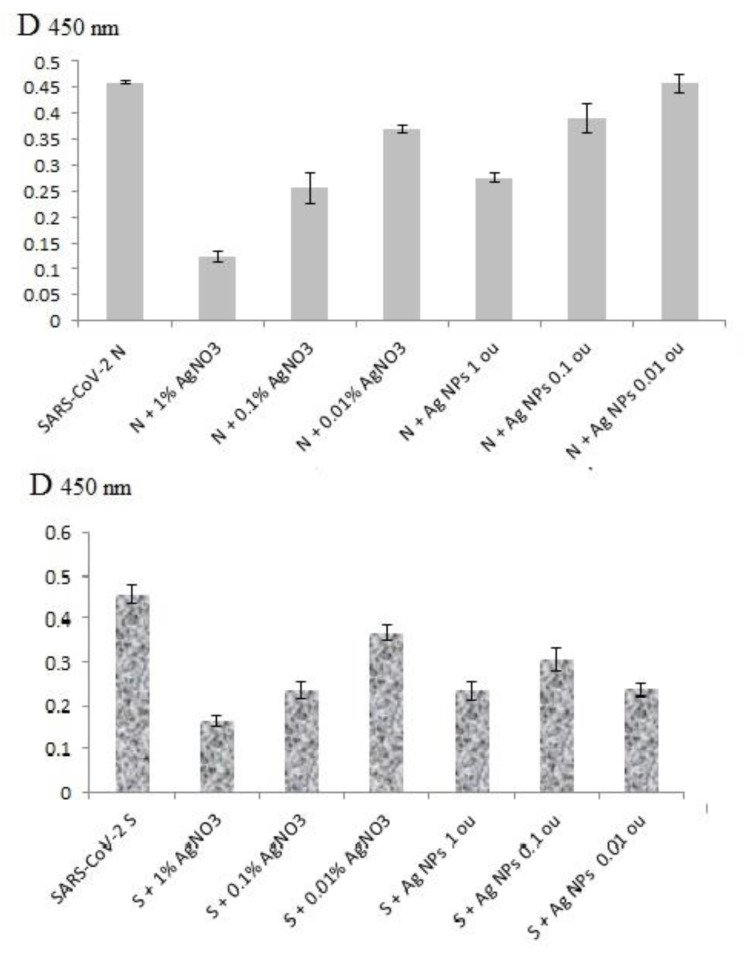
ELISA data for SARS-CoV-2 recombinant antigens N and S2 after their preliminary treatment with different concentrations of AgNO_3_ and Ag NPs.

**Table 1 viruses-14-00902-t001:** Cytotoxic concentrations (CC_50_, μg/mL) of Ag-containing materials for human cell lines.

Ag Nanomaterials	HEp-2	L41	HT-29
AgNO_3_	50 (31.75 Ag+) *	100 (63.49 Ag+)	100 (63.49 Ag+)
Ag-2S	5.3 (1.08 Ag+)	21.2 (4.32 Ag+)	15.9 (3.24 Ag+)
Ag NCs-IgG	7	10	7
nanoconjugates Ag NPs-IgG	5	15	5

Note: * CC_50_ (concentration that caused 50% cytotoxicity of eukaryotic cells) is shown as concentration of the studied substances AgNO_3_ (MW = 169.87)_,_ Ag-2S (MW = 530.05) and corresponding concentrations of Ag^+^ are in brackets. For Ag NCs-IgG CC_50_ for protein were 1 mg/mL and 10 mg/mL for HEp-2(HT-29) and L41 cell lines, respectively and CC_50_ (by metal content) for Ag NCs was calculated from protein CC_50_ on the base of preliminary mass spectroscopy data. For Ag NPs-IgG silver concentration was determined on the base of atomic absorption spectroscopy (AAS) after preliminary treatment of the nanocongugate with HNO_3_.

**Table 2 viruses-14-00902-t002:** Comparison of RNase activity of the silver nanomaterials at equal concentrations 0.1 mg/mL using RT^2^-PCR.

Nanosilver	MS2 RNA	MS2 Phage	SARS-CoV-2 RNA	SARS-CoV-2 RNA from COVID-19 Patient Blood
**AgNO_3_**	n/d *	2.0 **	0.9	0 ***
**Ag-2S**	n/d	7.5	n/d	1.6
**Ag NCs**	0	0	0	0
**Ag NPs**	3.7	0.9	1.8	0
**Ag NPs-BSA**	3.7	0	1.3	0
**Ag NPs-hIgG**	0	0	0	0

Notes: * n/d means that targeted RNA was not detected because of complete cleavage. ** average shift of threshold cycles of fluorescence (Ct) between sample after treatment with silver-containing materials and control untreated sample (ΔCt = Ct (sample) − Ct (control)). *** “O” corresponds to insignificant Ct shift <1 compared to control sample in the presence of sterile water.

**Table 3 viruses-14-00902-t003:** Unknown (X) Multi Analyte Profiling (xMAP) results for 17 inflammation biomarkers in HEp-2 cells in 2 days posttreatment. Normalization was carried out as a ratio of mean fluorescence intensity (MFI) for wells after incubation with the nanosilver to MFI of control intact HEp-2 cells.

	Th1				Th2								Th17	Others			
Inflammation Biomarkers	IFNγ	TNFα	IL-1β	IL-12(p70)	IL-2	IL-4	IL-5	IL-6	IL-7	IL-8	IL-10	IL-13	IL-17A	G-CSF	GM-CSF	MCP-1 (MCAP)	MIP-1β
**AgNO_3_**	1.03	1.15 ↑	1.38 ↑	0.70	0.74	0.84	0.88	0.85	0.70	3.71 ↑	0.78	0.81	0.97	1.12 ↑	0.92	1.19 ↑	1.17 ↑
**Ag NCs-BSA ***	0.93	0.78	0.63	0.82	0.54	0.78	0.83	1.05	0.91	0.93	0.79	0.95	0.62	0.76	0.88	0.94	0.68
**Ag NPs-BSA**	0.76	0.72	0.70	0.70	0.82	0.74	0.70	0.75	0.50	0.64	0.78	0.69	0.76	0.78	0.79	0.85	0.76
**Ag NPs-Fb**	0.72	0.65	1.00	0.72	0.78	0.75	0.82	0.75	0.55	1.00	0.77	0.83	0.84	1.00	0.83	1.15 ↑	0.97
**Ag NPs-IgG**	0.68	0.51	2.11 ↑	0.54	0.48	0.43	0.60	0.41	0.44	0.34	0.54	0.50	0.97	1.20 ↑	0.83	0.76	0.66

Note: * Ag NCs with red fluorescence (excitation at 340–540 nm; fluorescence emission maximum at 650 nm) (Appendix A).

## Data Availability

Original research data and any additional information will be available on request.

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
