# Peer review of "Targeting of Silver Cations, Silver-Cystine Complexes, Ag Nanoclusters, and Nanoparticles towards SARS-CoV-2 RNA and Recombinant Virion Proteins"

_viruses, 2022, doi:10.3390/v14050902_

Round 1

Reviewer 1 Report

In the manuscript “Molecular targeting of silver cations, complex with cysteine, nanoclusters and nanoparticles towards SARS-CoV-2 RNA and recombinant virion proteins”, Morozova et al. investigate the targeting and RNA-cleaving capacity of several silver-containing agents, and provide important information for antiviral application of silver-containing agents. The manuscript could be considered for acceptance after major revision:

(1) The title should be revised as “Targeting of silver cations, silver-cystein complex, silver nanoclusters and silver nanoparticles towards SARS-CoV-2 RNA and recombinant virion proteins” for avoiding confusion.

(2) What is the difference between the traditional RT-PCR and RT2-PCR the manuscript used? This should be introduced.

(3) In Table 1, AgNO3 is not nanomaterials. Thus the title “Ag nanomaterial” should be corrected. Furthermore, the readers could not recognize whether the cytotoxic concentrations (CC50, mg/ml) of the nanosilver indicate the concentration of Ag or the concentration of the agents (AgNO3, Ag-2S, etc.). It is suggested that the cytotoxic concentration of Ag element should be used for better comparison between the agents.

(4) Silver dissolution from the nanomaterials should be further investigated and the contribution of dissolved silver cation to RNA cleavage should be determined and discussed.

(5) The authors should test the effect of the silver-containing agents on immune cells, e.g., macrophages and neutrophils to better reflect their effect on immune response.

Author Response

Reviewer 1

In the manuscript “Molecular targeting of silver cations, complex with cysteine, nanoclusters and nanoparticles towards SARS-CoV-2 RNA and recombinant virion proteins”, Morozova et al. investigate the targeting and RNA-cleaving capacity of several silver-containing agents, and provide important information for antiviral application of silver-containing agents. The manuscript could be considered for acceptance after major revision:

  • The title should be revised as “Targeting of silver cations, silver-cystein complex, silver nanoclusters and silver nanoparticles towards SARS-CoV-2 RNA and recombinant virion proteins” for avoiding confusion.

Answer:

Former title “Molecular targeting …” was indeed inaccurate since nanoparticles and nanoconjugates are not molecules. Therefore, it was replaced as recommended.

  • What is the difference between the traditional RT-PCR and RT2-PCR the manuscript used? This should be introduced.

Answer:

Abbreviation (RT)2-PCR was used for reverse transcription with quantitative real time PCR as described at the beginning of the text in Abstract. Other abbreviation RT-PCR was not used in our manuscript and was mentioned only once in the title of the former reference #19 corresponding to the new reference number 20 of our revised reference list. It means reverse transcription and polymerase chain reaction without real time detection (by means of end point detection).

  • In Table 1, AgNO3 is not nanomaterials. Thus the title “Ag nanomaterial” should be corrected. Furthermore, the readers could not recognize whether the cytotoxic concentrations (CC50, mg/ml) of the nanosilver indicate the concentration of Ag or the concentration of the agents (AgNO3, Ag-2S, etc.). It is suggested that the cytotoxic concentration of Ag element should be used for better comparison between the agents.

Answer:

            The title of the revised Table 1 was changed to the following.

“Cytotoxic concentrations (CC50, mg/ml) of Ag-containing materials for human cell lines”.

The concentrations of Ag+ were added in brackets.

            Protein CC50 for Ag NC – IgG were 1 mg/ml and 10 mg/ml for HEp-2(HT-29) and L41 cell lines, respectively. CC50 (by metal content) for NC was calculated from protein CC50 on the base of preliminary mass spectroscopy data.  

For the quantitative determination of Ag concentration in silver nanoparticles and their nanocojugates with surface proteins atomic absorption spectroscopy (AAS) based on absorption of light by free metallic ions was used. Colloid solutions of Ag NP and nanoconjugates were preliminary treated with nitric acid in order to dissolve Ag+. Therefore, data in Table 2 correspond to Ag+ content in Ag NC-IgG and Ag NP – hIgG .

  • Silver dissolution from the nanomaterials should be further investigated and the contribution of dissolved silver cation to RNA cleavage should be determined and discussed.

Answer:

Silver dissolution from nanoparticles and corresponding nanoconjugates in water and physiological solution was previously studied in details (current references 24 and 25 (former ## 22, 23) and references therein). 

Morozova O.V., Klinov D.V. (2021) Nanosilver in biomedicine: advantages and restrictions. In book "Silver Micro-Nanoparticles - Properties, Synthesis, Characterization, and Applications," ISBN: 978-1-83968-660-3. DOI: 10.5772/intechopen.96331. https://www.intechopen.com/chapters/75544. ISBN: 978-1-83968-660-3. Print ISBN: 978-1-83968-659-7.  eBook (PDF) ISBN: 978-1-83968-661-0.

Morozova O.V. Silver Nanostructures: Limited Sensitivity of Detection, Toxicity and Anti-Inflammation Effects.  International Journal of Molecular Sciences 2021; 22(18):9928. DOI: 10.3390/ijms22189928.

Ag NP exhibit a “core−shell structure” with metallic silver in their central part surrounded by the surface oxide or sulfide layers as the outer shell (Li et al., 2012). Ag+ ions release in process of Ag NP dissolving. Replacement of Ag+ by electrolyte ions, potential formation of insoluble AgCl, subsequent catalyzed oxidative corrosion of Ag and further dissolution of surface layer of Ag2O take place (Li et al., 2012; Morozova et al., 2018). Oxidative dissolution of metallic Ag NP in the presence of an electron acceptor is catalyzed by nucleophilic reagents which change the chemical potential or Fermi level at the particle surface. This oxidation is controlled by the difference in the chemical potential between AgNP (with nucleophilic or stabilizing agents) and an electron acceptor. For uncoated silver nanoparticles, the oxidation of Ag(0) to Ag+ at the particle surface shifts the chemical potential of the particle to a more positive value, and if it approaches that of the electron acceptor (e.g., O2), oxidation ceases. The opposite shift in the potential occurs for metallic Ag NP with adsorbed nucleophiles (e.g., Cl or NO3), resulting in an increase in the oxidation of Ag(0) (Li et al., 2012). Therefore, Ag NP in the presence of ions and especially after addition of EDTA are not stable due to oxidation, dissolution and aggregation during a few hours. Nanosilver is evolving or ageing in contrast to dissolved Ag species (Reidy et al., 2013). However, our measurements using UV-visible spectroscopy, dynamic light scattering (DLS) and scanning electron microscopy (SEM) revealed that the citrate coated Ag NP remained stable colloid solutions in deionized water at room temperature for decades but not in the presence of ions. Citrate coated Ag NP are not stable in the presence of phosphate buffer solution (PBS) (0.01 M Na2HPO4/KH2PO4, 0.15 M NaCl/KCl) during 1 hour at room temperature due to replacement of Ag+ by Na+ and K+ ions (Li et al., 2012; Morozova et al., 2018). To prevent Ag NP dissolution and aggregation various surfactants and polymers are introduced during or after synthesis (Li et al., 2012). Coating layers may enhance electrostatic and steric repulsion. Adsorption of polymers or nonionic surfactants provides steric hindrances depending upon the thickness of the adsorbed layer (Li et al., 2012).

The NP stability depends on the affinity of coating molecules to the particle surface, repulsion from neighboring molecules, loss of chain entropy upon adsorption, and also nonspecific dipole interactions between the macromolecule, the solvent, and the surface. Protein corona protect Ag

NP from dissolution and aggregation. The nanoconjugates of the noble metal NP with proteins remain stable at +40C for several months (Morozova et al., 2018).

Li X., Lenhart J.J.,  Walker H.W. Aggregation kinetics and dissolution of coated silver nanoparticles. Langmuir. 2012;28(2):1095-1104. DOI:  10.1021/la202328n.

Morozova O.V., Volosneva O.N., Levchenko O.A., Barinov N.A. and Klinov D.V. Protein corona on gold and silver nanoparticles. Materials Science Forum. 2018; 936: 42-46. DOI: https://doi.org/10.4028/www.scientific.net/MSF.936.42.

Reidy B., Haase A., Luch A.,  Dawson K.A,  Lynch I. Mechanisms of Silver Nanoparticle Release, Transformation and Toxicity: A Critical Review of Current Knowledge and Recommendations for Future Studies and Applications. Materials (Basel) 2013;6(6):2295-2350. DOI:10.3390/ma6062295.

  • The authors should test the effect of the silver-containing agents on immune cells, e.g., macrophages and neutrophils to better reflect their effect on immune response.

Answer:

According to our previous data cytotoxic concentrations differ for various cell lines and tissue cultures (Ivleva E.A., Obraztsova E.A., Pavlova E.R., Morozova O.V., Ivanov D.G., Kononikhin A.S., Klinov D.V. Albumin-stabilized fluorescent metal nanoclusters: fabrication, physico-chemical properties and cytotoxicity. doi: 10.1016/j.matdes.2020.108771 License CC BY-NC-ND 4.0; Materials and Design. 2020;192:108771.  doi:10.1016/j.matdes.2020.108771). Anti-inflammation properties of Ag-containing materials analyzed by RT2-PCR, ELISA and xMAP for three human tissue cultures HEp-2, L41 and HT-29 in dynamics were similar. Table 3 shows only part of available results. Text of the section 3.4. Anti-inflammation properties of the nanosilver was revised. Unfortunately, currently we do not have cell lines of macrophages and neutrophils and hope for further research.

Reviewer 2 Report

Generally, the topic and contents of paper are quite interesting and highly relevant to prevalent issues posed by COVID-19, within the scope of viruses. Synthesis, and characterizations, their antiviral properties against SARS-CoV-2 coronavirus of various silver nanomaterials are convincing.  

Before its acceptance, the manuscript needs to be revised.

  1. Perhaps authors could provide some examples of antiviral potential of silver nanomaterials again SATS-CoV2? If possible, the mechanisms as well, at end of introduction? Brief explanation/summary for previous reports will also be helpful. Further elaboration on the cytotoxicity of silver nanomaterials as a limitation would be helpful.

  1. The TEM of Ag-2S should be included in Figure 1. Put them together for comparisons of Ag NC and Ag NP, and explain why TEM and AFM showed various sizes.

There was no mentioning of why AgNP and AgNO3 was selected for ELISA in Figure 3 among the different nanosilver? Is it based on ELISA results in figure 4?

  1. Figure S1-S3 are in the main text, can choose figure S1 in the main text and other two in the supplementary data.

A lot of typos throughout the manuscript, for example, line 189, “cysteine”; line 359 “intercellular”,  line 48, “ histogram”, Figure 1, 0.6 – 2; line 423, line 474, etc.

  1. The section of conclusion can be more elaborated, as in the results section.

Author Response

Reviewer 2

Generally, the topic and contents of paper are quite interesting and highly relevant to prevalent issues posed by COVID-19, within the scope of viruses. Synthesis, and characterizations, their antiviral properties against SARS-CoV-2 coronavirus of various silver nanomaterials are convincing.  

Before its acceptance, the manuscript needs to be revised.

  1. Perhaps authors could provide some examples of antiviral potential of silver nanomaterials again SATS-CoV2? If possible, the mechanisms as well, at end of introduction? Brief explanation/summary for previous reports will also be helpful. Further elaboration on the cytotoxicity of silver nanomaterials as a limitation would be helpful.

 Answer:

  Little is currently known about antiviral properties of the silver matters against coronaviruses (Du et al., 2018). Before COVID-19 pandemia Ag nanoclusters with glutathione were shown to inhibit other coronavirus RNA-dependent  RNA replication  and maturation (Du et al., 2018). Ag NP were shown to inhibit viral entry via disrupting viral integrity (Jeremiah et al., 2020). There is not a nanosilver among FDA-approved drugs (including remdesivir, ritonavir-boosted nirmatrelvir, sotrovimab and molnupiravir) as described at the end of the revised section “Introduction”. Recently published papers describing fluorescent Ag NC for biosensor-mediated coronavirus detection (Li et al., 2021) were added to the revised text.

To our knowledge neither small-molecular-weight Ag containing complex compound nor other artificial RNAses were analyzed against SARS-CoV-2 isolated RNA and virions. Comparison of different Ag based materials towards SARS-CoV-2 RNA and recombinant proteins was not earlier described.

Despite extensive research and already proven antiviral properties the nanosilver implementation in commercial antiviral drugs is limited (Munir et al., 2020; Kowalczyk et al., 2021; Morozova, 2021). Cytotoxicity is a major limitation for their use as antiviral agents in biomedical applications (Jeevanandam et al., 2022).  Multiple mechanisms of nanosilver action including three modes: ‘Trojan horse’, inductive and quantum-mechanical with possible collaborative effect that are responsible for antiviral and cytotoxic properties were described in our revised manuscript.

  1. The TEM of Ag-2S should be included in Figure 1. Put them together for comparisons of Ag NC and Ag NP, and explain why TEM and AFM showed various sizes.

There was no mentioning of why AgNP and AgNO3 was selected for ELISA in Figure 3 among the different nanosilver? Is it based on ELISA results in figure 4?

Answer:

Water-soluble silver (I) complex with cistine Li+[Ag+2Cys2-(OH-)2(NH3)2] (short name Ag-2S, C6H19Ag2S2LiN4O6, MW=530.05) had been earlier described in details (Tretyakov et al., 2007). This low-molecular-weight complex is hardly possible to detect by TEM even using uranyl acetate contrasting. Two Ag+ remain undetectable, therefore the TEM images were not shown. To detect the small molecules high resolution atomic force microscopy (AFM) with ultra-sharp carbon tips connected with silicon probe can be used (Obraztsova et al., 2019).  

 Obraztsova E.A.,  Basmanov D.V.,  Barinov N.A.,  Klinov D.V. Carbon Nanospikes: Synthesis, characterization and application for high resolution AFM Ultramicroscopy. 2019 Feb;197:11-15. PMID: 30447556 DOI: 10.1016/j.ultramic.2018.11.004.

            There are numerous tools that can be used for characterizing nanomaterials. Their results do not completely coincide because of sample preparation and physical conditions. Thus, both transmission and scanning electron microscopy are performed in deep vaccum whereas AFM is carried out in the air at normal atmospheric pressure. Therefore, sizes measured by electron microscopy can be less than those based on AFM. Besides that, TEM permits to determine diameters of 2-dimensional (2D) images whereas AFM allows to measure heights of nanomaterials (Revised Fig. 2). 

TEM is useful to distinguish silver materials (NC or NP) and surrounding protein envelopes. Thus, blue fluorescent Ag NC with IgG were in a range 0.6-2 nm (Fig. 1) and can be distinguished from surrounding less dense protein stabilizers and numerous aggregates and agglomerates by means of TEM only (Fig. 1). However, surface topography using AFM resulted in overestimations due to protein stabilizers (IgG) with known heights ~ 5 nm (Barinov et al., 2016) (Fig. 2) and did not permit to determine real size ranges of Ag NC (Fig. 1). On the contrary, AFM of the complex compound Ag-2S allowed us to compare diameters and heights of nanostructures and to reveal spherical molecules of 1-2 nm (Fig. 2). Ag NP and their nanoconjugates with proteins also significantly varied in diameters from 20 to 150 nm (Fig. 1) with polydispersity indexes 0.1¸0.2.

            Figure 3 shows a general scheme of our research with final results without explanations and preliminary data. The figure can be considered as the Graphycal Abstact. Really ELISA using SARS-CoV-2 recombinant proteins after treatment with all studied Ag-containing materials was performed (part of available data is shown on Fig. 4). Among five studied materials only two AgNO3 and Ag NP can damage antigenic determinants of SARS-CoV-2 recombinant proteins whereas three others Ag-2S, Ag NC-IgG and Ag NP-IgG did not change the binding of the coronaviral proteins with polyclonal antibodies and therefore may be excluded from further consideration (Fig. 4). Dependence of antigen-antibody interaction on AgNO3 and Ag NP concentrations is shown for both SARS-CoV-2 N and S recombinant proteins (next Figure 5) and part of available ELISA data are presented inside the scheme of Figure 3.

  1.  Figure S1-S3 are in the main text, can choose figure S1 in the main text and other two in the supplementary data.

Answer:

Physico-chemical properties of the studied Ag-containing materials are not the main goals of the manuscript that aimed at analysis of the SARS-CoV-2 targets for nanosilver with immunomodulation properties. Our revised manuscript already contains 5 multi-panel figures and 3 tables. Therefore, we believe that physico-chemical properties of Ag-2S (Supplementary Data Figure S1) and Ag NC with BSA (Supplementary Data Figure S2) and human IgG (Supplementary Data Figure S3) may be shown in Supplementary Data. 

A lot of typos throughout the manuscript, for example, line 189, “cysteine”; line 359 “intercellular”,  line 48, “ histogram”, Figure 1, 0.6 – 2; line 423, line 474, etc.

Answer:

            We are very grateful to the reviewer for attention to our research and thorough reading of the manuscript. Sorry for our spelling mistakes. Our revised text was carefully checked and our spelling mistakes were corrected.

            As to “histogram” different spelling is used in dictionaries (both histogram and hystogram). As one can see Word spell checking recommends “histogram”.

  1. The section of conclusion can be more elaborated, as in the results section.

Answer:

The section “Conclusion” was revised to more accurately match the results.

Reviewer 3 Report

This reviewer has the following suggestions for this manuscript.

1) The Abstract section is not precise enough and very hard to follow and thus must be rewritten.

2) The introduction section is also too long and should be switched to a more concise one with proper references cited.

3) Figure 1: why did the authors use different scale bars to indicate the size of NP? Size distribution from DLS (if there is) might be better. 0.6-(not ÷)2 nm? Same for line 382: 0.1-0.2

4) ALL figure legends should be extended to an informative style.

5) The data from table 1 are misleading, and this reviewer thinks that the cc50 for the listed nanomaterials is incorrect (e.g., 50, 100, 100). Moreover, the number (significant digit) used must be unified as xx.xx if possible.

6) The descriptions for the RNA cleavage section seem vague and must be summarized again. Currently, the overall flow is quite loose and not accessible clearly. The authors could use subsections to list the major findings in this part.

7) The data presentation must be improved and carefully verified before submission.

Author Response

Reviewer 3

This reviewer has the following suggestions for this manuscript.

  • The Abstract section is not precise enough and very hard to follow and thus must be rewritten.

Answer:

The Abstract was revised according to the recommendation.

  • The introduction section is also too long and should be switched to a more concise one with proper references cited.

Answer:

The “Introduction” section was also revised with addition of references describing antiviral properties of nanosilver against SARS-CoV-2 according to recommendations of all reviewers.

“Little is currently known about antiviral properties of the silver matters against coronaviruses. Antiviral activity of silver nanoclusters was assayed against the coronavirus species - porcine epidemic diarrhea virus (PEDV) (Du et al., 2018).  The fluorescent Ag NC were shown to inhibit viral RNA-dependent RNA replication and  PEDV maturation (Du et al., 2018). Ag NP were studied with similar coronaviruses in vitro and in vivo that along with toxicological data allowed to suggest COVID-19 possible treatment (Pilaquinga et al., 2021).  Ag NP of diameter ~10 nm were effective in inhibiting extracellular SARS-CoV-2 at concentrations 1-10 ppm while cytotoxic effect was observed at concentrations of 20 ppm. Ag NP were shown to inhibit viral entry via disrupting viral integrity (Jeremiah et al., 2020). Fluorescent Ag NC can be also used for coronavirus detection (Li et al., 2021). To our knowledge neither small-molecular-weight Ag containing complex compounds nor other artificial RNases were analyzed against SARS-CoV-2 isolated RNA and virions. Comparison of different Ag based materials towards SARS-CoV-2 RNA and recombinant proteins was not earlier described”.

  • Figure 1: why did the authors use different scale bars to indicate the size of NP? Size distribution from DLS (if there is) might be better. 0.6-(not ÷)2 nm? Same for line 382: 0.1-0.2

Answer:

There are numerous tools that can be used for characterizing nanomaterials. Their results do not completely coincide because of various sample preparation and physical conditions. Thus, both transmission and scanning electron microscopy are performed in deep vaccum whereas AFM is carried out in the air at normal atmospheric pressure. TEM permit to determine diamemetrs whereas AFM can measure heights of nanomaterials (Revised Fig. 2). 

To our knowledge hydrodynamic radii determined by using dynamic light scattering (DLS) may also result in overestimations of sizes in comparison with other available methods. To reveals silver core in complex silver-containing nanomaterials TEM is recommended as “gold standard”. While it is best to have multiple, orthogonal characterization techniques, there is an international consensus for transmission electron microscopy (TEM) being the “gold standard” technique for nanomaterial characterization. The European Food Safety Authority requires two methods to characterize size distribution of nanomaterials in foods, one of which must be TEM and other may be chosen (Qin et al., 2022). To corroborate the TEM findings, Atomic Force Microscopy (AFM) showing height images and size distribution histogram was often employed (Qin et al., 2022).

Qin N., Hemmes P., Mitchen K. Characterization of the Silver Nanoparticles in the Sovereign Silver® and Argentyn 23® , Bio-Active Silver Hydrosol™ Products. International Journal of Nanomedicine 2022:17 983–986. doi:10.2147/IJN.S355084.

On the base of TEM, AFM and DLS sizes of the studied Ag-containing materials vary in ranges shown on Figure 1 and revised Figure 2 (with additional histograms of size distribution). Blue fluorescent Ag NC were in a range 0.6-2 nm (Fig. 1) resulted from different numbers of recovered Ag atoms in each NC and can be distinguished from surrounding less dense protein stabilizers and numerous aggregates and agglomerates by means of TEM (Fig. 1). Surface topography of Ag NC using AFM resulted in overestimations due to protein stabilizers (IgG) with known heights ~ 5 nm (Barinov et al., 2016) (Fig. 2) and did not permit to determine real size ranges of Ag NC (Fig. 1). On the contrary, AFM of the complex compound Ag-2S allowed us to compare diameters and heights and to reveal spherical molecules of 1-2 nm (Fig. 2). Ag NP and their nanoconjugates with proteins also significantly varied in diameters from 20 to 150 nm (Fig. 1) with polydispersity indexes 0.1¸0.2.

  • ALL figure legends should be extended to an informative style.

Answer:

Done.

  • The data from table 1 are misleading, and this reviewer thinks that the cc50 for the listed nanomaterials is incorrect (e.g., 50, 100, 100). Moreover, the number (significant digit) used must be unified as xx.xx if possible.

Answer:

The Table 1 was revised according to recommendations of two reviewers.

CC50 (concentrations that caused 50% cytotoxicity of eukaryotic cells) were previously   shown as concentrations of the studied substances AgNO3 (MW=169.87) and Ag-2S (MW=530.05) but after revision the corresponding concentrations of Ag+ were added in brackets. For Ag NC-IgG CC50 for protein were 1 mg/ml and 10 mg/ml for HEp-2(HT-29) and L41 cell lines, respectively, and CC50 (by metal content) for Ag NC was calculated from protein cytotoxic concentrations on the base of preliminary MALDI-TOF mass spectrometry data (not shown in this manuscript but in preparation for other paper). For Ag NP–IgG silver concentration was determined on the base of atomic absorption spectroscopy (AAS) after preliminary treatment of the nanocongugates with H3NO3.

  • The descriptions for the RNA cleavage section seem vague and must be summarized again. Currently, the overall flow is quite loose and not accessible clearly. The authors could use subsections to list the major findings in this part.

Answer:

Subsections were added in the revised section. RT2-PCR data were summarized.

  • The data presentation must be improved and carefully verified before submission.

Answer:

Figure 2 was changed with AFM bar scale and two histograms showing height distribution. Table 2 was also revised with additional calculations of Ag content in each analyzed compound. Data presentation was carefully verified.

Round 2

Reviewer 1 Report

The authors have carefully revised the manuscript. It is now acceptable for publication in Viruses.

Reviewer 3 Report

This reviewer has no further comments.